# Dynamic academic networking concept and its links with English language skills and research productivity–non-Anglophone context

**Anna L. Wieczorek**[1]*, **Maciej Mitręga**[2], **Vojtěch Spáčil**[3]

**1** University of Bielsko-Biala, Bielsko-Biala, Poland, **2** University of Economics in Katowice, Katowice, Poland, **3** VSB—Technical University of Ostrava, Ostrava, Czech Republic

* awieczorek@ath.bielsko.pl

**Data Availability Statement:** All relevant data are within the paper and its Supporting Information files.

## Abstract

Although the Science of Team Science or SciTS has already provided substantial evidence for research collaboration positive links to scientific productivity, much less is known about such links with broadly defined academic networking, especially with regard to the dilemma about forms of academic networking that may help individual scholars in handling risks and dynamics inherent in academic connections. This study uses cross-disciplinary theoretical insights to conceptualize "dynamic academic networking" as a distinct collaboration-related phenomenon that is theoretically linked with research productivity on the one hand, and with English language skills on the other, especially in the context of non-Anglophone academic systems. The study combines survey-based data and Scopus-based data to test two main hypothesized connections while controlling for the potential effects of other factors, e.g. home faculty research connections and faculty-industry professional connections. The research results provide support for the structural model which is also interpreted in terms of dynamic networking being valid concept in relation to further development of SciTS.

## Introduction

Scientific productivity manifested by a large number of publications is said to become a holy Grail of many academics [1]. Although expectations of PhD candidates towards academic enrolment are high and job at university is still perceived by many as a "dream job" [2], scientific work in reality has become very stressful, because scholars must face increasing demands in various areas such as writing papers, applying for external funds, orienting teaching at practical competences, and administration [3]. The global trend towards increased number of graduates from doctoral studies, particularly in business education, contrasts with the worsening situation on the demand side, where universities have problems with retaining qualified staff members, facing their own resource constraints and increased competitiveness [4]. Although the increased competitiveness and diffusion of so-called "publish or perish culture"

**Funding:** The research presented in this paper was financed by the National Science Centre (NCN) in Poland according to the decision no. DEC-2012/05/E/HS4/02216.

**Competing interests:** The authors have declared that no competing interests exist.

is noticed worldwide, the regional paths bring different contexts into how this culture can be aligned with specific institutional environments of academic systems in some regions [5].

One of the widely acknowledged mechanisms of increasing research productivity is research collaboration, the phenomenon that has many facets and its definition has gone through some evolution in the last two decades [6–8]. The importance of this mechanism was discussed in so many studies that the new interdisciplinary research area was proclaimed, namely the Science of Team Science or SciTS [9]. Although SciTS provides well documented insights into how research teams are built and what factors help in effective team management, much less is known about tools and tactics that individuals may use while networking, specifically to mitigate collaboration risks and dealing with academic ties instability [6, 10]. There is also a call to integrate within SciTS various theories, and methods from across disciplines, because research collaboration is a clearly interdisciplinary research phenomenon [9]. This study corresponds with the gaps identified in the literature by proposing a "dynamic academic networking" concept (DN) that is conceptually distinct to the traditional, narrow concepts of research collaboration, but refers to wide spectrum of academic interactions which may be all perceived as important academic tasks [11, 12]. Specifically, following self-interest theories of organizational networks [13] and dynamic capabilities view in strategy research [14], we propose DN as routinized patterns of scholar behaviour that are oriented at learning and productivity development through academic relationships at various stages of their development. We also use our knowledge on the networking processes in non-Anglophone academic systems to hypothesise English language skills as an antecedent of DN in such context. Our study addressed the call for cross-theoretical and cross-disciplinary approach to understand better links between research collaboration and research productivity by using insights not only from SciTS, but also from organizational and strategy research and linguistic studies. Last but not least, our research model with DN as a focal construct is validated through two data collections, one using self-reported survey measures and the other combining some of these survey measures with Scopus based data on scholars' productivity. Both of these studies are limited to the data gathered about scholars specialized in business science employed in Poland as a representative of non-Anglophone countries.

This paper is structured as follows: first we review the SciTS literature to provide distinction between various forms of academic collaboration and conceptualize dynamic academic networking as "fresh" research angle. Then we introduce two hypotheses related to DN as a focal constructs to position DN in a nomological network [15–17] and we briefly present the post-communist academic systems, mainly using Poland as an example, to build reader's understanding of the context of non-Anglophone academic systems. Then we present research methods and research results providing the support for the hypothesized model. Finally, we discuss research findings, provide their practical implications and identify limitations as well as further research avenues.

## Literature review

### Academic collaboration and academic networking in Science of Team Science

Some forms of academic collaboration already were considered in the research of academic productivity many years ago [18], however studies devoted solely to this topic appeared later and since 90ties they have been growing visibly. Particularly, in the 2000s, so-called Science of Team Science (SciTS), i.e. distinct cross-disciplinary research area focusing on team science was acknowledged [9]. The main premise of SciTS is that the innovative research comes from diversity of resources that are brought into research projects and academic teamwork creates a

complex platform for managing such diversity [19]. The empirical evidence for the significant impact of academic collaboration on scholars' productivity is rich and it illustrates that academic collaboration may increase productivity in both terms: quantity (e.g. the number of publications) and quality (e.g. citations, journal impact factor) of academic output. Researchers receiving peer support and sharing ideas, expertise and resources provided by collaboration in peer networks are, in view of [20], to have higher productivity rates than their more isolated colleagues. This evidence comes mainly from most developed countries such as USA, Australia, Canada, Spain and Italy. Additionally, the majority of the researches were conducted in the context of a single scientific discipline, which means that this evidence is rather fragmented [21]. Some studies that used multi-disciplinary perspective provided mixed evidence suggesting that the importance of academic collaboration varies across fields of science in the same country (e.g. [22]).

The collaborative projects in academia were primarily popular in the STEM field (Science, Technology, Engineering and Mathematics) [12], but they were more and more practised in social sciences, mainly due to the trend towards organizing research through programmes, centres and networks initiated and funded by national research councils and international agencies [12]. This rise of collaborative projects goes hand in hand with growing internationalization of research. It was noticed that the percentage of ISI scientific works that were co-authored by scholars from various countries grew dynamically from 8.7 percent in 1990 to 15.6 in 2000 [18]. While academic teamwork and co-authorship are still less popular in social sciences than in other disciplines, the co-authorship trend is similar like in "big science" [10, 23]. The team social science appears to be a very complex, fascinating and largely neglected research phenomenon [8]. Logically, in comparison to other disciplines team social science is also very risky, because social science research methods are less standardized and the researchers' cultural backgrounds are more meaningful due to interpretative epistemology bringing more potential for team conflicts.

The SciTS has progressed substantially in the last 2 decades which is visible through the recent attempts to map its development [6, 9, 10]. Particularly, our knowledge is much advanced with regard to how scholars select partners [e.g. 24, 25] what team composition works [e.g. 26], what team processes help in effective team functioning [e.g. 27, 28] and how institutional resource availability impacts on initiating and managing research collaboration [e.g. 19, 29]. However, SciTS is fragmented with relation to research disciplines and many things seem to be unsolved. First of all, the SciTS is dominated by measuring academic collaboration through co-authorship which depreciates all forms of academic collaboration that did not finish in joint publications [6, 9]. Such approach puts aside all meaningful interactions between scholars such as exchanging ideas, sharing research techniques and research data that may, but not necessarily must, result in publishing together [21]. While sharply delimiting the boundaries of what is academic collaboration and what it is not might be problematic, especially in terms of when it starts and finishes [7], the academic collaboration might be perceived broadly as all interactions between at least two scholars that help in sharing of meaning and executing academic tasks [8]. Recently [6], defined academic collaboration in even wider terms as "*social processes whereby human beings pool their human capital for the objective of producing knowledge*" (p. 3). Such perspective gives appropriate flexibility for studying team science in their various forms and connecting these forms with research effectiveness at both levels: team effectiveness, e.g. publishing together and individual effectiveness, e.g. publishing individually using insights from collaboration. In fact, in comparison to teamwork in typical commercial organizations [30], academic collaboration at all its stages, including deciding whether to initiate it at all, is much more based on decisions of individuals being "*able to act as free agents*" ([9] p. 542). Identifying trends and gaps in SciTS [10], suggests that one of the

most promising avenues is looking at the collaboration from the perspective of individual scholars involved and impacting that various forms of collaboration made on them. Therefore, SciTS may benefit not only by utilizing insights from intra-organizational teamwork management literature [31, 32], but it may also use inspirations from inter-organizational relationship management literature, where business entities are perceived as being less or more capable in collaborating and applying collaboration strategies to retrieve relational rents [33–35]. Similarly scholars may apply various tactics towards their professional relationships and leverage their performance this way.

Secondly, SciTS is recently found as presenting too rosy picture of academic collaboration, while there is a need to prepare researchers for facing collaboration costs effectively [10]. In fact, research collaboration brings many problems which if not properly managed, may outweigh the cooperation benefits. As academic collaboration often happens between actors very different in terms of their work experience and their resources, there is a real threat of exploitation and free riding of a stronger against the weaker one, while the weaker does the vast majority of work [36]. In turn, this threat is especially important with regard to social sciences [37]. This is not a trivial problem, because the diversity as the main premise of effective team science somehow implies the power asymmetry between cooperating scholars and the equal contribution and credit related to team science cannot be ever delivered in real life and, perhaps should be not even facilitated on an institutional level. The mentoring is identified as one of typical styles of collaboration usually associated with senior scholars cooperating with junior scholars and students [38] and this style can happen always when a more experienced scholar interacts with a less experienced one. For example, such cooperation style is typical between scholars employed at the same academic positions but having very different publication records. It is also typical for cooperation between scientist from countries at very different levels of their development [39]. In general, brokering which refers to some scholars occupying the most resourceful and beneficial position in academic networks is rather a typical phenomenon [40, 41], however from the perspective of individual scholars that are not "stars" yet, there is a need to apply some ways to mitigate the risk of being exploited, or in more general terms, "*to mitigate the costs of collaboration*" ([10] p. 88). Logically, the scholar should manage collaboration inputs with collaboration outputs appropriately, especially in the longer time frame. They should also aim at gradually strengthening position towards being less dependent, e.g. managing their own research teams. Even without conscious attempts of one partner towards exploiting the other partner, the team science often becomes problematic due to unclear governance of research projects; the epistemic and ethical responsibility becomes too blurred and nobody feels fully responsible for the teamwork [42]. Thus, using the knowledge on business relationship cycles [43–45], all academic collaborations may be treated as instable occurrences, i.e. accepting that some of them never mature enough and some of them already lost their attractiveness and should be not maintained any more. Such approach towards collaboration seems relevant in social sciences, where the tangible resources, e.g. laboratories, dedicated to given academic teams are relatively small and scholars may engage in and disengage from various subsequent collaborations that develop slowly.

This work assumes that one of the ways to overcome current weaknesses of SciTS is to focus on academic "networking" instead of just academic "collaboration" from the perspective of a given scholar that functions in the "publish or perish" realm. Functioning in this context enforces proving usefulness of a scholar through publication productivity, but it also motivates for strengthening one's own assets and building career. Here we follow [12] that proposes networking as one of major tasks of the academic researcher nowadays and defines networking as a "communication pattern" which comprises exchanging ideas, results and information between scholars via various channels that may, but not necessarily has to, involve

collaborative research and collaborative publishing. Academic collaboration, in turn, takes more organized forms and usually results in joint publishing [12].

Although the literature offers fuzzy boundaries for using the term research/academic/scientific collaboration [7, 8] that could potentially involve communication pattern we write here about, we think that making distinction between these two terms is important, because in practice the academic collaboration is mainly measured through co-authorship [9, 10], thus it is associated by SciTS scholars rather with advanced forms of research teamwork. Such twist of a team science towards more developed acts of cooperation is visible even on a conceptual level. For example, although [8] proposes wide definition of scientific collaboration that equates collaboration with just interaction between scientists, the very same definition suggests that such interaction "*facilitates the sharing of meaning and completion of tasks with respect to a mutually shared*, *superordinate goal*" (p. 645). As we have already illustrated such "mutuality" and "shared meaning" does not need to happen in academic interactions, so we prefer to use the term "networking" to tackle the variety of interactions that happen in academic relationships and networks from the perspective of a focal actor. In turn, such focal actor being resource constrained has also large degree of autonomy with regard to "who" to network with and "how" to network, especially in social sciences that we focus on. Our understanding of networking as important academic task is aligned with [21] that emphasizes flexible interactions with peers in academia as the platform for developing new ideas and [21] illustrating empirically the differences between forms of academic cooperation across disciplines and concludes that informal interactions are significant for all disciplines and should be more facilitated in policy making. Evidence was provided that such informal academic communication might be more important than formal cooperation and co-authorship [46], especially in case of developing countries.

Corresponding with the idea of individual scholars being relatively autonomous in their networking decisions, especially in social sciences [8, 10], we are inspired by dynamic capabilities approach in strategy research [14, 47] and self-interest theories of organizational networks [13]. Dynamic capabilities approach suggests developing some specific capabilities that enable anticipating the opportunities and threats sensed in the environment [14]. In such strategy, the dynamic capabilities take a of form either some simple radical decisions, i.e. in rapidly changing situations, or routinized processes to handle environmental dynamics effectively, i.e. in moderately dynamic environments [48, 49]. In line with this approach, we perceive academic networking as generally happening in dynamic environment, i.e. we presume that the various forms of academic interaction, including collaborations are unstable and managing existing teamwork although very important is not the only task that the focal scholar must focus on if he/she wants to leverage his/her own productivity and success. Therefore, among various theories of organizational networks, we correspond here with self-interest theories that explain actors' behaviour in organizational networks through their orientation at exploiting opportunities and cost minimizing rather than mutuality or group identities [13]. Consequently, we propose the concept of **Dynamic Networking (DN)** and we define it as *routinized focal scholar patterns of dynamic interaction with other scholars that happen at various stages of collaboration cycle and are oriented at learning from others and purposively shaping the network of a focal scholar*. We understand DN not in binary terms, i.e. either one scholar having a DN or not, but instead we understand DN as a continuous feature of every scholar, with analogy to the concept of individual personality trait [50]. Similarly as there are not just agreeable or not agreeable people but all people are characterized by some level of agreeableness, scholars cannot be easily divided into the cohort of dynamic networkers and stable networkers. For example, one person might be quite active in socializing at conferences and successful in initiating some new academic contacts and that feature would suggest advanced DN, however the same

person might have problems in critically evaluating existing research collaborations and that feature, in turn, would suggest limited DN. Low inclination towards downsizing in some existing relationships and moving attention towards new promising ones may come from various reasons including scholar personality, investments in current relationships, the lack of other opportunities, etc.

Contributing to the SciTS with proposing DN as a meaningful concept in the context of other collaboration concepts widely known in the literature such as "colocation" or "university-industry collaboration" [8] cannot happen automatically because these other concepts have proven their explanatory power through many studies before [9]. So far, we believe we have provided some conceptual rationale for treating networking in general and DN particularly as the concepts referring to important academic activities which are broader than the concept of scientific collaboration and encompass the dynamic nature of team science. However, introducing DN into SciTS must happen through establishing its nomological validity [15–17], thus proving important position of this concept in the nomological network of SciTS, i.e. its relations with some other constructs that function as its antecedents and consequences. That is why this study proposes the model, where DN functions as a central concept and it is related with two other factors: scholar's publication productivity as DN consequence and scholar's English language skills as DN antecedent. At the same time, this study controls for the relationships between DN and other potentially related concepts that were previously discussed in the literature. We will elaborate on these controls further in the relevant subsection, i.e. "Control variables".

## English skills and dynamic networking

Although publishing in English language became almost the sole mode of effective research results' dissemination in today's academia, especially since "publish or perish" culture also dominated academic systems of developing countries [51], the command of English language is not equally spread across countries and this inequality negatively impacts careers of scholars from non-Anglophone context [52, 53]. Indeed, in comparison to other languages English might be treated as a "global language" [54, 55] which is spoken by1,121 million speakers (native and non-native ones). English is also recognized as an official language in almost 1/3 countries in the world [55], and generally, the whole Impact Factor publication world is English-speaking due to the fact that the majority of highly-impacted journals are English-speaking and this means that a non-Anglophone author must have high reading (in order to do a proper literature review) and writing (in order to deliver high-quality papers) skills [3]. It clearly shows that English is not only a global language in terms of numbers of people speaking it in general, but is also is a global language of science. Still, however, in many countries, especially outside Europe, this language is spoken by less than 10 percent of population (e.g. Brazil, China, Chile and Argentina [56]). There are also several EU countries, including post-communist countries (Latvia, Slovakia, Poland, Romania, Czech Republic, Bulgaria and Hungary) where less than 1/3 population speaks English [57]. Additionally, in all post-communist countries there is a substantial disproportion of English languages skills between people at various age groups, because in these countries since the end of WW2 and the beginning of 90. last century, it was common to teach the Russian language as a foreign language at schools as all of these countries were Soviet Union satellite countries in that period. Thus, in these countries, older people usually have a low knowledge of English language, which crosses all industries and professions in the society. Here we call all countries with relatively low level of English language non-Anglophone countries.

Many inhabitants of non-Anglophone countries not only have worse command of English, but they also perceive the world differently which is caused by general cultural differences and

also by the language they speak and by absence of some notions in their native language which, in turn, may make some phenomena difficult to understand for those people. Sapir–Whorf hypothesis [58, 59] explains differences in human cognition [60] and emotion [61] through differences in human language. In the context of academic interactions, it means that the way the scholar constructs his/her worldview is based on the language the scholar acquired and uses. Extrapolating this mechanism to the concept of DN, it could be that acquiring English language skills makes non-Anglophone scholars not only more capable of English-language based communication with other scholars, but also it may change their approach to interaction with other scholars. Specifically, as English language is perceived a language rich in pragmatics [62], the acquisition of English language may make a non-Anglophone scholar more pragmatic in interaction with other scholars, which enable DN and indirectly, it has positive effect on scholar's productivity. Furthermore, taking into consideration that we build our understanding of the world through the lens of our native language, the lack of some words in a given language influences the way people think and act as its native users. We may take the notion of *networking* as an example of it. In Polish language there is no equivalent of the English word *networking* and many people simply do not know the word/concept at all. The notion is known by some business people, scientists or researchers who study this phenomenon, but they speak and write about it borrowing the English term, so one can assume they do not have a preunderstanding of the concept–those who do not do research on networking may even not know what networking really entails and as a result do not have proper abilities to network with others. In turn, a term *collaboration* which is closely connected with *networking*, but of course is not synonymous, has negative connotations for some Poles as it in their minds this notion may refer mainly to the traitors of the nation collaborating with Germans at the times of II World War (such WWII practices were called *collaboration*, *"kolaboracja"* in Polish). This can make some scholars associate the concept of collaboration with unethical cooperation practices (like e.g. ghost writing) and limits their understanding of the wide spectrum of networking actions that are available, i.e. initiating new connections with scholars for mutual learning.

Knowing that the command of English language is still not developed enough in quite many countries in the world, it is easy to imagine that "*non-Anglophone scholars are linguistically disadvantaged relative to native-speaking academics when it comes to publication in English*". [63], but it is also very easy to imagine that English skills are an important predictor for dynamic networking by scholars embedded in all non-Anglophone academic systems. In general, in such context the English language skills might be an intermediate step for getting published in quality English journals, because language skills are important for initiating and maintaining contacts with experienced, well published academics, while in turn, these effective academics offer some less experienced scholars many learning opportunities, including possibility to learn new research methods, understand research gaps better and practice "English for Research Publication Purposes" [52]. The review of 39 empirical studies that investigated multilingual scholars' participation in academic communities suggests that language difficulties isolate scholars from non-Anglophone countries from core/global academic communities [64]. Curry [53] demonstrated that for many scholars from southern and central Europe the individual linguistic competence is usually not sufficient for publishing in quality English-medium journals, but it is very important for getting access and participating in academic networks that facilitate such publication results. The study suggests that English language enables non-Anglophone scholars to communicate with better published colleagues outside their closest network and these contacts many times transform into joint research projects and co-authorship [53]. In another work [65] suggest that for scholars for whom English is not their first language, English language skills are an important element of so-called "social practice"

that is crucial for successful publishing and includes such activities as participating in international projects and leading conferences, choosing an appropriate journal to publish in, responding to reviewers, etc. Noteworthy, some elements of this "social practice" overlap with dynamic networking, e.g. participating in leading conferences. The study of narratives by successful scholars in business science from Poland and other Central European Countries [3, 66] suggests that improving English language skills was crucial for these scholars to get connected with mainstream science on both: personal and methodological level and it was a very difficult challenge, especially for those scholars that did not learn English at initial levels of their education. Quite intuitively, as dynamic networking concept assumes fostering pragmatic interactions, i.e. target-oriented interactions between focal scholar and other scholars, this may be very difficult to achieve if the focal scholar does not exhibit good language-based communication skills, because such dynamic interactions demand having good understanding of what is happening in these interactions and reacting appropriately if these interactions go to the wrong direction, i.e. towards being clearly exploited, towards unprofessional connections, etc. Thus, we hypothesize:

H1. *The higher level of English language skills is positively liked to higher level of dynamic networking of Scholars from non-Anglophone context.*

## Dynamic networking and academic productivity

As already mentioned SciTS provided vast evidence for the positive connection between various forms of research collaboration and research productivity [8–10], however the connection between dynamic networking (DN) and research productivity was not explored yet. In this study we introduce DN as a new construct brining a new angle to the team science. Therefore, justification of the hypothesized connection between DN and productivity could be built on combing the understanding of proposed DN concept with our knowledge of processes happing in academic relationship networks. Again, being aware of other theories explaining organizational networks, e.g. mutual interest and collective action theories, proximity theories, in this study we follow self-interest theory of organizational networks that explains networking behaviour through individual actors' instrumental orientation at collaborating with other people [13]. Therefore, we associate DN with scholars' interactions with their colleagues that are oriented at maximizing benefits and minimizing costs of networking. Following prior studies on dynamic networking in business networks [33–35] here we propose that DN takes a form of managing academic relationships at various stages of their development: systematic actions to initiate some promising relationships (1), analysing existing relationships in terms of their current attractiveness with regard to publishing potential (2). We propose that both of these elements have some positive impact on focal scholars' productivity.

Initiating new contacts with appropriately selected scholars leverages scholar research productivity mainly because such routinized attempts to enlarge academic network assure access to "fresh blood" connections with variety of experienced scholars and it is an important mechanism for increasing one's competences one the one hand, and it is a necessary step towards building new successful collaborations in the long-run, on the other. Such premature academic relationships, i.e. these that do not result in co-authorship yet, are almost neglected in SciTS [21], however they can help the focal scholar to learn many things useful in further, either individual or joint, publishing. In fact, various forms of academic collaboration are viewed as very effective platforms for knowledge transfer [67]. According to [8], when applying the perspective of an individual scholar this refers to academic collaborations at various stages of their

advancement: "*Scientists may recognize this in the formulation stage (learning can be a motivation to initiate a collaboration)*" (p. 666). Additionally, knowledge and skills that may be learned through networking go beyond any existing research projects, because the scholar may learn research methods, project management skills and advanced English academic writing [8]. As an example, one can imagine two scholars that initiated contacts at the conference and exchanged opinions on each other's work-in-progress works, then they also continued ideas exchange on a remote basis. While they also had in mind some joint research opportunities, such collaboration never happened because of some time-restrictions or perspective mismatch that was revealed through further interactions. Even though co-authorships did not result from these interactions, they could have a very important impact on at least one of these scholar's productivity, because learning could happen through them as well as other processes important for publishing in journals, e.g. exchanging contacts to other colleagues or becoming more confident that the given research area is worth investigating. We also expect that systematic attempts of a focal scholar to enlarge academic network through dynamic networking increase the likelihood that the focal scholar's network would be rich in diverse contacts, e.g. with people from other disciplines or other countries, while such diversity of resources is treated as the main mechanism of a team science to be extraordinarily productive [19]. Systematically initiating new contacts may build more collaboration options for a focal scholar to select from, instead of exploiting only these network resources that are already available.

Analysing current academic relationships in terms of their changing attractiveness is also important for the productivity of a focal scholar, because current collaborations are exposed to many threats and they are also cyclical in nature, they can be rather never productive in an indefinite way. As already mentioned in this paper, there are many things that may happen in academic relationships that may outweigh potential benefits of collaboration such as free riding and exploitation problems [36, 37] or being too dependent on academic "gatekeepers", i.e. these that have strongest networks and easiest access to resources [10]. The problems with sharing work load, blurred responsibility for task execution and dysfunctional conflicts may likely emerge as a natural trajectory in some academic relationships, especially when "*scientists have misconceptions regarding the resources, including time, required to conduct various research tasks.*" ([8], p. 658). Therefore, DN in terms of periodic analysis of a current portfolio of academic relationships might be crucial to prevent the focal scholar from "getting stuck" in some personally intensive, long-lasting connections and free resources for some new promising relations. We argue that such an approach could be especially beneficial for a focal scholar in social sciences, where in comparison to "big science disciplines", the tangible investments in existing projects are not big, so there are usually no substantial "switching costs", e.g. dedicated infrastructure, connected with downsizing or even ending existing projects and moving attention to some new promising collaborators.

Thus, we hypothesize: H2. *The higher level dynamic networking of scholars is positively linked to their research productivity in non-Anglophone context.*

## Post-communist business science in Poland as a non-Anglophone research context

Polish academia like in case of many other CEE members (Central and Eastern Europe) of EU is transforming from communist-type of academia [68], where the contact with global science was prevented by the "iron curtain" of so-called satellite countries of Soviet Union. The existence of this geopolitical barrier for half of a century, resulted in Polish scholars being disadvantaged in terms of their understanding of worldwide research methods, most popular research topics and, obviously, also in terms of their English language capabilities. This is

especially true with regard to senior scholars, who learned Russian at school, i.e. before Eastern Block collapsed at the end of 80s. This is a serious barrier for their research output, because quality publishing demands fluent English, not only for writing papers but also for getting access to most updated scientific sources. Additionally, foreign langue is necessary for conducting large scale research projects (e.g. EU grants) which are welcome in quality journals and usually demand some form of international cooperation. Even National Centre of Science (NCN) as the main domestic institution providing research grants in Poland demands nowadays preparing the whole application in English language, as this application, after first expert check, is evaluated by experienced foreign scholars.

National Centre of Science (NCN) offers special track for these research projects that are prepared in cooperation with foreign scholars. However, international cooperation in today's academia demands appropriate level of language skills [69, 70] and this means developing English language skills in both terms: English writing (for preparing papers for top-tier journals, jointly applying for funds) and English speaking (for daily project teamwork, meeting at conferences). While in Western Anglophone countries, older researchers are usually more productive and more active in networking [22, 71], in Poland younger scholars are usually more productive [68] and it is connected with the fact that they on average older scholars do not know English well. Vast majority of Polish publications are still outside of major international datasets [68], which can be caused by not being ready to be published in English language recognisable journals.

In the last two decades Polish academic system has gradually become more "open" (e.g. increasing participation in international conferences and top-tier journals), however Polish language has remained main professional language for Polish scholars (e.g. the tradition of PhD thesis writing in Polish language) and individual research achievements have been more rewarded than teamwork academic output like co-authorship [72], contrary to Western countries where teams dominate scientific endeavours [73]. Since the collapse of the communist system in Poland at the beginning of 90., Polish scholars, especially those specialized in business-related areas, have got involved in part-time jobs at fatly developing sector of private universities and have established business connections with dynamically developing private sector mainly for getting extra-money (e.g. cooperating as business consultants). Although some of these professional connections could be utilized for scientific purposes, this potential is usually not used as low social capital in Poland [74, 75], which is necessary if business practitioners are supposed to openly share their business data with business researchers. On the other hand, the working ties built with the other scholars help in scientific productivity, if they allow for combining complementary skills and resources (e.g. [12, 21]). For example, in business science in Poland there is a popular practice of inviting some statisticians to the team, which provide expertise needed for sophisticated analysis of empirical data. Similarly, some more experienced scholars, especially those that are well published and have some working ties with top scholars in the field are likely invited by other scholars for joint research projects. This way, some more experienced Polish scholars can function as structural holes for their colleagues at the same home faculty [76].

In general, cooperation with scholars that bring very different resources and competencies, especially well established foreign researchers can be very fruitful for an average business scholar employed at Polish university. Such cooperation assures better integration of a scholar with the mainstream science (i.e. access to cutting-edge methodologies and newest topics) and, which translates into stronger pro-publishing motivation, including publications in quality journals [77]. However, exploiting such professional ties demands first some active networking which allows for going beyond the established social embeddedness and building new extra-organizational connections. Without networking skills scholars from the transforming

Polish academia are usually left on their own in the process of finding research collaborators, because scholars, unlike their Western colleagues [69], usually do not get access to network connections through their supervisors as those supervisors are even more oriented at "prior communist academic culture" and usually do not have established working ties with scholars established in international academia. This mechanism was illustrated with regard to all Visegrad Countries (Poland, Czech Republic, Hungary and Slovakia) through in-depth interviews and international survey among young academics [78].

It should be also mentioned that similarly to some other post-communist countries, Polish academic system, including business science or social sciences in large terms, experienced dynamic changes in the last two decades. It is connected with institutional changes that have for example increased the importance of impact factor publications in the university domestic accreditation system. Although the scholars from humanities and social sciences were traditionally less published, i.e. in terms of publication numbers and, especially, less represented in international databases as compared to representatives of Polish scientist from other disciplines, e.g. medicine or informatics, analysing publication activities of Polish scholars from social sciences and humanities in recent years [79], found that the percentage of non-publishing scholars has decreased enormously and when it comes to number of publications Polish scholars in this disciplines become similarly productive as scholars from more advanced West European countries. Among all analysed disciplines related to social sciences, the scholars specialized in business were found as most productive. Nevertheless, this shift towards increased productivity is visible mostly in terms of publication quantity, whereas publication quality in terms of impact factor publications is still relatively low.

The pace of changes of Polish academic system, similarly to other post-communist systems of CEE (Central and Eastern Europe) has substantially progressed in the last decade. Analysing the publication patterns of non-English-speaking European countries: Czech Republic, Denmark, Finland, Flanders (Belgium), Norway, Poland, Slovakia, and Slovenia [80], found that in the period between 2011 and 2014 the publication patterns were stable in West European and Nordic countries, whereas CEE countries experienced considerable changes towards more and more emphasis in English language publications in international journals. For Poland as the biggest CEE country the trend continues with regard to impact factor publications which is largely based on the strong impact of policy instruments with regard to country's national research evaluation system, academic promotion procedures and competitive grants [81]. In October 2018 the new Law on Higher Education and Science was established in Poland (also called *Science Constitution 2.0* to emphasize the depth of institutional changes) which centralized leadership of the university authorities [82] and at the same time imposed much more demanding measures of scientific productivity based on publication outputs in Scopus database. Specifically, the new ranking of publications in journals was mostly based on SNIP (The Source-Normalized Impact per Paper) indicator which is a relatively new journal metric by Elsevier's Scopus [83]. The SNIP was selected because this indicator acknowledges citation potential related to differences in topicality across scientific fields. Therefore, it may be concluded that in contrast to previous periods, nowadays the research productivity of business scholars in Poland is evaluated mostly through their publications registered in Scopus. This institutional change motivated Polish scholars to publish more and more in highly ranked journals, so it may be assumed that the trend identified for the period 2011–2016 [80, 81] received some additional impetus. It translates into even more challenging situation for the older generation of Polish scholars who started their careers in communist period, where publications in Polish language were the main pattern of dissemination of their research results.

## Control variables

The research model presented at Fig 1 comprises DN as a focal construct and its hypothesized antecedent (English language skills) and consequence (research productivity) which constitutes the basic nomological set in this study (the baseline model). However, to make sure that DN is really a construct that allows for looking at SciTS from a relatively new angle, we have included several important control variables into the model. Firstly, the home network corresponds with the academic colocation/proximity concept [8, 24] from the perspective of a focal scholar and refers to the scholar's quality of professional relationships that take place within home faculty academic ties, i.e. work contacts that have been built between focal scholars and their peers at home university. In turn, Corporate network corresponds with University-industry collaboration concept [8] from the perspective of an individual scholar and refers to professional relationships between focal scholar and business practitioners. This specific cross-industry collaboration was perceived as a driver of publication productivity in prior research, because it increases the innovativeness of academic research [84, 85], however as explained before in this study, it is treated as a control variable only, due to specific the institutional context of our research (see previous section). The Fig 1 presents our hypothesized research model. In next section we will briefly explain the interrelation between DN, English language skills and publication productivity and then we will present the features of post-communist social science system in Poland that provides non-Anglophone context for studying DN. Additionally, the model controls for some other variables not related to the forms of academic collaboration but offering alternative explanations to dependent variables under consideration that were considered in prior research on academic productivity [86, 87]. Specifically, the model controls for the impact of Teaching load (i.e. the number of hours devoted to teaching that potentially limits scholar's research orientation), Resource support (perceived back-office resource support for research projects), Character (perceived conscientiousness as a personality trait helpful in preparing and executing research projects), Culture (perceived extent to which university culture is oriented at productivity that motivates scholars for making extraordinary research efforts), Academic rank (the position that given scholar is employed at that

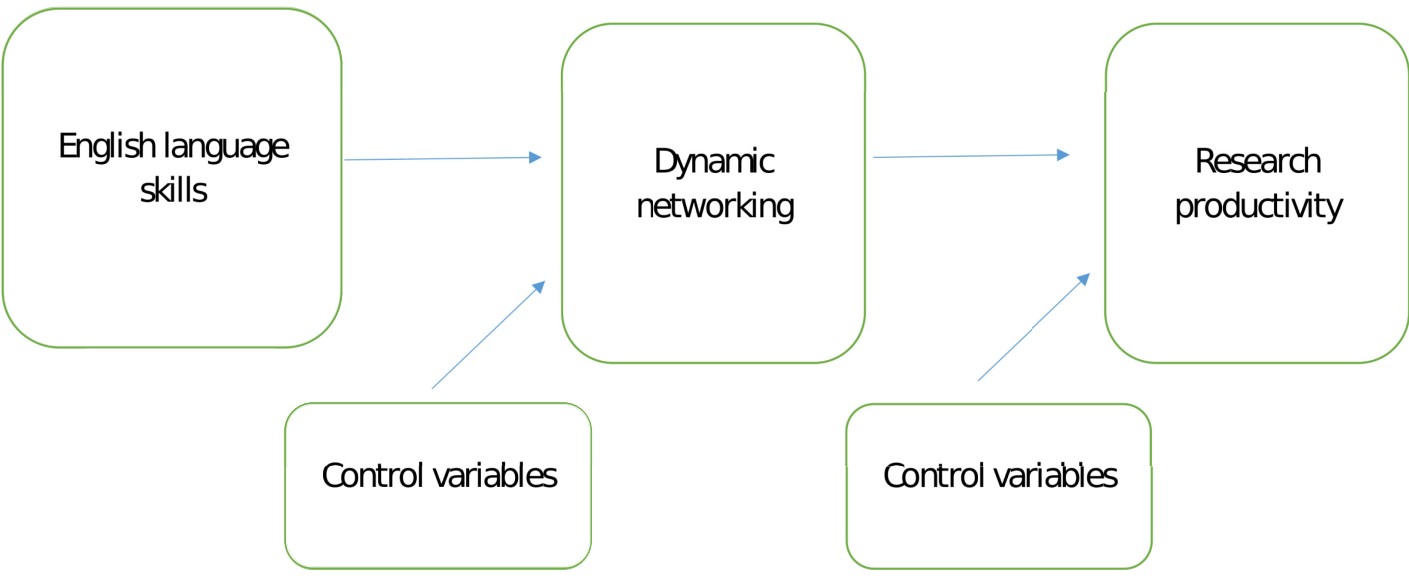

**Fig 1. The research model with DN as a focal construct in nomological set.**

provides better access to research resources), Employment time (i.e. time of employment at university) and Age.

## Methods

### Measures development

This study follows [46], who suggest bibliometric techniques as very problematic for investigating scholars from less developed countries, because such scholars are not well represented in international scientific databases. Focusing only on bibliometric sources such as Scopus or Google Scholar was found as irrelevant, because such databases do not provide the way to measure scholars' networking activities except co-authorship patterns. Similarly, we were not able to use any publicly available sources for measuring scholars language skills as available proxies, e.g. publishing in English were found as too general. We also wanted to control for the influence of some additional variables, e.g. conscientiousness or perceived academic culture at the university, so we have used self-report for all research variables similarly to [46] for vast majority of our research variables. However, to cross-validate research results provided by using only self-reported survey results, we have conducted ex-post robustness test, where we have utilized Scopus-based publication scores as a measure of research productivity for those scholars who participated in our survey, for whom we were able to find appropriate Scopus researcher's profiles. We will refer to the study conducted only with the use of self-reported data as a study 1 and the study combining survey data with Scopus data will be called study 2.

In study 1 the research productivity was measured in the way aligned with specific features of transforming academic systems in CEE [78]. Specifically, CEE scholars are very underrepresented in top-tier journals and, on the other hand, they publish massively in non-impact factor journals and, to large extent, in Polish-medium journals and Polish-medium research monographs [68, 88]. We also followed [46] who suggested that in the context of less developed countries papers published in local and foreign journals both have some value from the perspective of academic career. To purify measurement from lowest quality publications we have clearly asked informants to indicate number of publications in "blind-review journals" with regard to last two years. Limiting perspective to output from last 2 years was done to control for the cumulative publication effect related to informants' age differences.

The survey measure of academic networking was inspired by research [89] which proposed item pool for networking capability in the context of inter-firm cooperation. We adapted these items to the context of academic networking on interpersonal level. The measures of the existing professional ties, namely Home network and Corporate network, were adapted from [90]: we distinguished between different types of academic partners and we incorporated self-reported measures of network quality complementing measures of network size. This method of measuring existing academic network was previously used by another research conducted in all 4 Visegrad countries illustrating that in case of all of these countries connections built by scholars are much more intensive within the boundaries of their home faculties than outside their universities [78]. Finally, English language skills were measured reflectively using one item referring to English language speaking and the other item referring to English language writing.

To measure the control variables we generally used single items or multi-items adapted from prior studies in the same area. Specifically, we adapted items provided by [86] and then replicated by [87] to measure teaching load and resource support. We used one of items proposed in previous researches [91, 92] to measure informants' character, specifically informants' conscientiousness as a personality variable. All other control variables; Productivity Culture, Employment time, Academic rank and Age, were generally measured as simple self-reporting

scales. The gender was measured as dichotomous biological gender (men vs. women) and introduced to the analysis as a dummy variable. All items that we have used in our survey are presented in the **S1 Appendix**.

In study 2 we have re-used the vast majority of survey measures for research variables with one exception which is that research productivity was measured through number of surveyed scholars' publications tracked in Scopus database. Specifically, we used Author search service of Scopus database, which allows for tracing how many publications registered in Scopus were published by a given scholar in a given calendar year. As study 2 was conducted 6 years later than study 1, to keep it comparable with publication measure used in study 1 we have focused on publications coming from the last 2 years only. Precisely, we used publication data visible for the period between 1 Jan 2018 and 17 July 2020.

## Research sample

Study 1 uses survey data gathered in April 2014 from senior scholars employed at universities in Poland and specializing in business-related fields, e.g. management, marketing and business logistics, human resource management, etc. Due to the absence of a relevant sampling frame, we used non-random sample of senior scholars, i.e. scholars at the age of 40 or older, because we wanted to focus only on scholars who started their primary education in the communist era, when Russian language was obligatorily taught at schools as a foreign language and it is rather a burden for them, i.e. in relation to other younger scholars in Poland exposed early to English language communication. We assured that the sample is diversified in terms of rank these universities had in relevant domestic ranking of business education and the university type, i.e. either business-focused university or general university type. Specifically, we included all publicly owned universities specialized solely in business, i.e. "akademie ekonomiczne" (higher schools of economics) that received highest ranks in terms of business education in Poland. Kozminski University was included as the biggest and most prestigious private university offering business education. We also included universities of general type that focus on business education only among teaching other disciplines and we assured that here we have covered universities of the highest rank, i.e. University of Warsaw as well as universities with relatively low rank with regard to business education, e.g. University of Szczecin, University of Zielona Góra, University of Warmia and Mazury, Częstochowa University of Technology and Silesian University of Technology. To increase the effectiveness of informants' selection we also allowed our informants to send the survey link to their peers from other institutions and in that case we asked informants to select only the most productive scholars from their social network. The surveyed universities' structure presented in the S1 Appendix suggests that around 1/3 of all informants participating in the survey were employed at universities of a relatively low rank in business education according to a well-known domestic ranking: *ranking. perspektywy.pl*. The publication results declared by informants position the surveyed population as a quite typical population of business scholars in Poland—concluded from comparing publication scores of survey informants with publication scores revealed in another larger scale study of business scholars in Poland [66].

We selected informants qualified as senior scholars through age filter question, i.e. only scholars that declared being 40 years old or older were selected for the analysis. Some questionnaires needed to be eliminated due to being provided by younger scholars, the lack of answers to main questions and "rapid clickers" which resulted in keeping 198 questionnaires from senior scholars for further analyses. Although people (i.e. survey informants) were involved in the research, the approval of any institutional review board (i.e. relevant ethics committee) before the study began was found as not necessary due to the fact that the survey was

anonymous and the research did not assume testing any form of physical or psychological treatment with regard to people involved in research. The research project was reviewed and approved by the National Science Centre in Poland which became the financing institution within the contract no. UMO-2012/05/E/HS4/02216.

In study 2 that was conducted in 2020 we have re-used survey data gathered in 2014 and combined the data with additional data about publication scores of the investigated informants that was acquired through Scopus. In line with the prior research suggesting that bibliometric sources are problematic for less developed countries, because such scholars are not well represented in international scientific databases [46], we have managed to get access to much smaller sample size than in study 1. Specifically, we retrieved relevant Scopus data for only 73 scholars participating in survey 1. Although, this sample is only a fraction from the initial database, it serves well as ex-post cross-validation test. Additionally, including Scopus data for the publication score in the period 1 Jan 2018 and 17 July 2020 extends the evidence used to test the hypothesized model from a cross-sectional study into a longitudinal study. Logically, routinized dynamic networking practices measured in the survey in 2014 should have long-term effects on scholars' publication scores, especially in social science, where academic collaborations take usually much longer from their beginning until the publication results [8]. The survey used in study 1 was anonymous, the participants however could reveal their e-mail address if they didn't mind not staying anonymous. Because of some e-mail addresses revealed we were able to identify some participants and check their Scopus records in study 2.

### Testing measurement quality

Before estimating the hypothesized links between our constructs, we first tested the measurement in terms of its reliability and validity. We used SmartPls 3.0. and our results show meeting all standard thresholds (see in the Table 1) except Cronbach's alpha for Resource Support amounting to 0.651, which is acceptable assuming the exploratory character of this research project [93, 94]. In case of Productivity measure we followed the suggestion by [95]—we first assured via bootstrapping that both indicators of productivity have significant path coefficients. Next we also tested variable inflation factors (VIF) for formative indicators and for these both indicators they were below conservative threshold of 3.3 indicating the collinearity is not an issue.

Finally, we controlled for heterotrait-monotrait ratio of correlations (HTMT) as the latest discriminant validity test suggested by [95] for PLS-SEM and all HTMT were below the suggested (conservative) threshold value of 0.85 (see in Table 2).

**Table 1. Construct reliability and convergent validity.**

|  | Cronbach's Alpha | rho_A | Composite Reliability | Average Variance Extracted (AVE) |
|---|---|---|---|---|
| Academic rank | Single item | Single item | Single item | Single item |
| Age | Single item | Single item | Single item | Single item |
| Character | Single item | Single item | Single item | Single item |
| Corporate network | 0.914 | 1.016 | 0.958 | 0.918 |
| Culture | Single item | Single item | Single item | Single item |
| Employment | 0.887 | 1.026 | 0.944 | 0.894 |
| English Skills | 0.960 | 0.963 | 0.980 | 0.961 |
| Home network | 0.861 | 0.867 | 0.935 | 0.878 |
| Dynamic networking | 0.860 | 1.003 | 0.894 | 0.629 |
| Productivity | Formative measure | Formative measure | Formative measure | Formative measure |
| Resource Support | 0.651 | 0.718 | 0.847 | 0.735 |
| Teaching load | Single item | Single item | Single item | Single item |

**Table 2. Construct discriminant validity by Heterotrait-Monotrait ratio (HTMT).**

|  | Academic rank | A | Ch | CN | C | E | ES | HN | DN | RS |
|---|---|---|---|---|---|---|---|---|---|---|
| Age (A) | 0.550 | | | | | | | | | |
| Character (Ch) | 0.014 | 0.081 | | | | | | | | |
| Corpotate network (CN) | 0.051 | 0.040 | 0.127 | | | | | | | |
| Culture (C) | 0.094 | 0.057 | 0.052 | 0.044 | | | | | | |
| Employment (E) | 0.617 | 0.718 | 0.073 | 0.065 | 0.056 | | | | | |
| English Skills (ES) | 0.013 | 0.086 | 0.128 | 0.294 | 0.035 | 0.088 | | | | |
| Home network (HN) | 0.026 | 0.024 | 0.010 | 0.268 | 0.138 | 0.081 | 0.184 | | | |
| Dynamic networking (DN) | 0.037 | 0.066 | 0.055 | 0.390 | 0.170 | 0.036 | 0.218 | 0.596 | | |
| Resource Support (RS) | 0.125 | 0.052 | 0.080 | 0.231 | 0.095 | 0.096 | 0.218 | 0.323 | 0.180 | |
| Teaching load | 0.184 | 0.212 | 0.121 | 0.119 | 0.005 | 0.138 | 0.047 | 0.156 | 0.075 | 0.183 |

## Results

To estimate the hypothesized links between variables we used partial least squares path modeling, which is also known as PLS-SEM and it is an alternative to covariance-based SEM (CB-SEM) that experienced exponential growth in terms of its applications in highly ranked journals in business research since the release of free available SmartPLS 2 software in 2005 [96]. PLS-SEM has recently become used in other areas of social science such as psychology [97, 98] and outside social sciences, in biology [e.g. 99] and medicine e.g. [100]. We decided to use PL-SEM technique instead other techniques, e.g. regression, CB-SEM, due to few reasons. First of all, our research model contains mediation effect, i.e. DN is mediator between English skills and research productivity, and in general such effects cannot be modelled using traditional regression techniques. Secondly, as we used survey data in our analysis (only survey data in study 1 and mix of survey data and Scopus data in survey 2) the distribution of our data was to large extent non-normal which suggested and PLS-SEM allows for using this kind of data, while other techniques do not allow for that [101]. Thirdly, in this research we introduce Dynamic networking as a new concept, our approach is to large extent exploratory or it is an extension to the existing theories related to organizational networks and dynamic capabilities, and in such instances PL-SEM is preferred over more conservative approaches such as CB-SEM [95]. Fourthly, as our sample, especially in study 2 is relatively small (n = 73) the use of PL-SEM is recommended and it "*is a good approximation of CB-SEM results*" [101 p. 144]. Last but not least, DN as well as several other antecedents in our model were conceptualized as latent variables–"reality" that cannot be directly observed but it is inferred from other variables that are observed. Specifically, DN is proposed here as a reflective latent variable which assumes that its measured indicators (5 Likert scales) are a sample of all possible indicators of DN latent constructs, i.e. dropping one indicator would usually not change the meaning of the construct [102]. Such approach to measurement is possible in PLS-SEM, while other techniques assume usually formative measurement, i.e. each indicator represents a dimension of meaning of the latent variable and dropping one dimension/indicator changes the meaning of the latent variable [102].

The hypothesized links between variables were estimated using the SmartPLS 3.0 software package [103]. Following [93], bootstrapping was used to assess the path coefficients' significance. The number of bootstrap samples was 5,000, with Bias-Corrected and Accelerated (BCa) Bootstrap as the most stable estimation method. Table 3 presents the results of the estimation of the structural model including control variables. None of the coefficients related to control variables, including Home network and Corporate Network, appeared to be

**Table 3. PLS-SEM path coefficients–study 1 (survey data, n = 198).**

| | Original Sample (O) | Sample Mean (M) | Standard Deviation (STDEV) | T Statistics (|O/STDEV|) | P Values |
|---|---|---|---|---|---|
| Academic rank -> Dynamic networking | 0.007 | 0.014 | 0.098 | 0.069 | 0.945 |
| Academic rank -> Productivity | -0.042 | -0.022 | 0.105 | 0.396 | 0.692 |
| Age -> Dynamic networking | -0.021 | -0.010 | 0.111 | 0.189 | 0.850 |
| Age -> Productivity | -0.027 | -0.018 | 0.113 | 0.242 | 0.809 |
| Character -> Dynamic networking | -0.040 | -0.040 | 0.073 | 0.552 | 0.581 |
| Character -> Productivity | 0.045 | 0.042 | 0.073 | 0.625 | 0.532 |
| Corpotate network -> Productivity | -0.104 | -0.093 | 0.089 | 1.166 | 0.244 |
| Culture -> Dynamic networking | 0.148 | 0.148 | 0.078 | 1.912 | 0.056 |
| Culture -> Productivity | -0.075 | -0.075 | 0.069 | 1.089 | 0.276 |
| Dynamic networking -> Productivity | 0.296 | 0.284 | 0.091 | 3.239 | 0.001 |
| Employment -> Dynamic networking | 0.035 | 0.010 | 0.129 | 0.276 | 0.783 |
| Employment -> Productivity | 0.126 | 0.092 | 0.122 | 1.031 | 0.303 |
| English Skills -> Dynamic networking | 0.173 | 0.173 | 0.084 | 2.055 | 0.040 |
| English Skills -> Productivity | 0.019 | 0.010 | 0.073 | 0.258 | 0.797 |
| Home network -> Productivity | 0.069 | 0.079 | 0.092 | 0.751 | 0.453 |
| Resource Support -> Dynamic networking | 0.109 | 0.124 | 0.079 | 1.380 | 0.168 |
| Resource Support -> Productivity | 0.041 | 0.045 | 0.078 | 0.529 | 0.597 |
| Teaching load -> Dynamic networking | -0.059 | -0.055 | 0.074 | 0.789 | 0.430 |
| Teaching load -> Productivity | 0.078 | 0.076 | 0.073 | 1.068 | 0.286 |
| Gender (men) -> Dynamic networking | -0.044 | -0.050 | 0.079 | 0.565 | 0.572 |
| Gender -> Productivity | -0.096 | -0.097 | 0.075 | 1.284 | 0.199 |

statistically significant (p>0.05), but the path coefficients related to dynamic academic networking is significant at conservative level (p<0.05), similarly, the path related to English language skills (p<0.05), thus providing support for hypotheses 1 and 2.

In ex-post robustness test study 2 we followed the same procedure with the use of the SmartPLS 3.0 software package [103], however the research results are slightly different due to the smaller sample size (n = 73) and due to using new data source for publication scores, i.e. Scopus data. In the analysis in study 2 we concentrated on estimation of hypothesized paths related to new productivity measure, that is why Table 4 presents the results of an estimation of the baseline model (hypotheses 1 & 2) and the estimated path coefficients for the links

**Table 4. PLS-SEM path coefficients–study 2 (survey & Scopus data, n = 73).**

| | Original Sample (O) | Sample Mean (M) | Standard Deviation (STDEV) | T Statistics (|O/STDEV|) | P Values |
|---|---|---|---|---|---|
| Academic rank -> Scopus_18–20 | 0.006 | 0.020 | 0.119 | 0.048 | 0.962 |
| Age -> Scopus_18–20 | 0.012 | 0.025 | 0.222 | 0.054 | 0.957 |
| Character -> Scopus_18–20 | 0.125 | 0.133 | 0.090 | 1.393 | 0.164 |
| Corpotate network -> Scopus_18–20 | 0.007 | -0.027 | 0.191 | 0.037 | 0.970 |
| Culture -> Scopus_18–20 | -0.062 | -0.054 | 0.101 | 0.608 | 0.544 |
| Employment -> Scopus_18–20 | 0.118 | 0.131 | 0.167 | 0.709 | 0.478 |
| English Skills -> Networking | 0.348 | 0.373 | 0.102 | 3.421 | 0.001 |
| English Skills -> Scopus_18–20 | -0.079 | -0.089 | 0.116 | 0.687 | 0.493 |
| Home network -> Scopus_18–20 | -0.005 | 0.023 | 0.144 | 0.033 | 0.974 |
| Networking -> Scopus_18–20 | 0.231 | 0.247 | 0.117 | 1.969 | 0.049 |
| Resource Support -> Scopus_18–20 | 0.172 | 0.125 | 0.259 | 0.663 | 0.508 |
| Teaching load -> Scopus_18–20 | -0.046 | -0.006 | 0.224 | 0.207 | 0.836 |

between control variables and research productivity. To sum up, study 2 provides the additional evidence for the hypothesized connections between English skills and DN on the one side and DN and research productivity on the other as baseline connections appear to be statistically significant.

## Discussion

The study contributes to SciTS literature by proposing the concept of dynamic networking (DN) that complements the various forms of academic collaboration that were already widely discussed, e.g. colocation teamworks, university-industry teamwork [8] and is nomologically valid, especially with relation to individual researcher's productivity. As SciTS is recently being criticized for looking too much at advanced and successful forms of interactions between academicians [9, 10], our study addressed these limitations by focusing on "networking" rather than "collaboration" or "teamwork". Reviewing the SciTS literature we provided the conceptual delimitation between networking and other forms of collaboration between scholars, positioning networking as routinized communication behaviour that sometimes takes a form of, but is not limited to, collaborative projects. Networking encompasses non-collaboration meaningful interactions, such as providing feedback, sharing ideas, which are very important from the perspective of a development of a given scholar. Indeed, if these interactions were not meaningful there would be no reason to organize scientific conferences, especially "work in progress papers" sessions which are very useful for scientific projects to be co-developed even through such co-creation usually does not transform into further enlarged co-authorship. Focus on networking is also well aligned with currently observable growth of academic social media like *academia.edu* or *researchgate.net*, where scholars discuss various research topics without necessity to meet physically and beyond any formal boundaries, including boundaries of being formally engaged within the same research project.

A scholar with advanced DN abilities behaves in a pragmatic way at all stages of academic relationships cycle, i.e. makes attempts to enlarge social network and critically analyses existing collaborations, so DN concept addresses also the other limitation acknowledged with regard to SciTS, which is that SciTS does not provide the individual scholars with the tools that could be useful to mitigate costs of collaboration [10, 36]. Obviously, some of these costs could be managed by the factors that were widely discussed in SciTS like partners selection and team composition e.g. [24–26] and team management practices e.g. [27, 28]. However following self-interest theory of organizational networks [13] and dynamic capabilities view of strategy [14], team composition and team management do not seem enough to prevent from disruptive elements of being embedded in academic network. As functionality of all academic ties is time restricted, some ties got never mature and some lose their attractiveness. We fully agree here with [6] that: "*Research collaborations, even useful ones, sometimes go down blind alleys. Researchable phenomena are inherently unpredictable, otherwise there would be no need for the research*" (p.3). Additionally, some collaboration threats like exploitation of one scholar by another one are not avoidable, because scholars, including these occupying powerful network positions use academic networking in a very instrumental way. Therefore, DN could be a very effective strategy to prevent the focal scholar from this relationship dynamics. Specifically, DN enables focal scholar to be more selective and flexible with regard to academic connections, so even if some existing collaborations appear not to be successful, applied DN assures that the focal scholar still has access to some other connections to focus on and to potentially develop into advanced collaborative projects or just learn from them to develop self-managed research projects.

Our study positions DN as an important SciTS concept with regard to non-Anglophone academic systems and it provides empirical evidence that DN is leveraged in Polish post-

communist academic realm through English-language skills, specifically in the group of senior business scholars employed at various universities in Poland. In general, this result confirms prior studies, where English language was found as an antecedent of engagement of non-Anglophone scholars into various forms of social practise in international academia, including international collaboration and taking into consideration the fact that it works as mechanism for making a non-Anglophone scholar more successful [53, 65]. However, these previous studies were mainly rather qualitative, so they did not allow for testing this link on larger group of researchers. The contribution of our study is also in providing the evidence that language skills are not only vital for advanced forms of academic collaboration, like joint research projects and co-writing, which is pretty obvious as far as English-language publications are concerned, but in demonstrating that English language skills help with specific pragmatic approach to networking reflected in DN concept. We argue that our research goes beyond a trivial conclusion that every person's networking must be based on appropriate language skills, because networking is the specific form of communication. We interpret our findings through two theories that are well known in language studies. Since our dataset concerns only senior scholars (age 40+), the importance of language skills for their networking may be related to classical critical period hypothesis in second language acquisition research, which postulates that people that did not learn language before certain age, i.e. puberty, find it more difficult to learn this language at later stages of their lifetime [104, 105]. Thus, in case of older scholars that usually did not have a chance to acquire English language skills at initial stages of education, the developed English language skills might be a strategic asset because they are rare in their age group and they may leverage their self-confidence which is logically essential in academic networking, as such networking assumes presenting qualities while meeting new scholars and being assertive in existing collaborations. Furthermore, bearing in mind the hypothesis by Whorf and Sapir [58, 59, 61] which states that people perceive the world through their native language, we can see that 40+ Polish scholars usually not speaking English well often do not have deep understanding of what networking and/or collaboration entails and thus cannot fully develop their potential.

Our research focuses on DN as a specific collaboration-related behaviour, however it also controls for the effect of some other collaboration forms, i.e. colocation (home network) and university-industry connections (corporate network), which in contrast to prior research [8, 9] do not appear to be significant predictors of research productivity in the investigated context of senior business scholars in post-communist Poland. The unimportance of corporate network This study enriches our knowledge on the connection between collaboration and productivity in academia by decomposing "the black box" of academic networking and proposing "the dynamic networking" (Networking) as a framework that reflects some behavioural routines undertaken by the focal scholar to build, enlarge and revise their professional network. Our study provides the evidence that such Networking contributes to academic productivity of senior scholars specialized in business and employed in Polish academic institutions, while controlling for the influence of other factors that were previously considered in productivity studies, e.g. academic rank, university culture and resource support. However, in contrast to other studies in this area conducted mainly with regard to scholars from most developed countries focusing on general link between teamwork and academic productivity disciplines [20, 46], this study suggests that not all forms of academic cooperation leverage productivity of senior Polish scholars. This way our research complements prior researches [17, 47] that postulated "strategic approach" to academic networking, in which focal actor builds network selectively.

The form of professional ties that is clearly found in this research as not useful resource for scientific productivity is corporate ties, i.e. connection built by the focal scholars with business

practitioners. This could be surprising in relation to business scientific discipline, because in general such contacts can potentially bring access to valuable research data and even inspirations about most up-to-date research topics. However, this result is reasonable if interpreted in the specific context of Polish transforming academia. Since the collapse of the communist system in Poland at the beginning of 90., Polish scholars, especially those specialized in business-related areas and these that hold PhD already in 90., i.e. currently senior scholars, have established business connections with dynamically developing private sector mainly for getting extra-money (e.g. cooperating as business consultants). As a consequence, such external ties with business were not necessarily aimed at scientific output, but they can be very effective in leveraging scholars' financial status, because university positions are not necessarily well paid. Our research results also show the importance of the proximity/colocation based relationships [8, 24] built by Polish business scholars. Although, our research does not question such local ties as more easily built ones (i.e. they may more likely happen as a platform for research collaboration), it suggests that they have no value for the productivity measures, i.e. publication scores. This result may be interpreted either through theories of SciTS or though specific features of our research context. The home faculty connections may not provide the adequate diversity of resources and skills, e.g. interdisciplinary teams, that is one of the main premises of effective team science [9, 19]. On the other hand, previous research on the motives and pattern of academic networking in post-communist countries in business science [3, 66] demonstrated that local collaborations are developed by scholars non only purely for scientific purposes but also as a safeguarding mechanism for their positions at home faculties. Additionally, the same research demonstrated that scholars tend to benefit especially from scientific contacts with experienced, well published colleagues s, that are frequently employed at foreign academic institutions. Thus, local peer collaborations, similarly to corporate connections do not have the impact on publication productivity, especially while their influence on productivity is compared with the influence of other factors, i.e. in estimation of our baseline model enhanced by the impact of all control variables.

Summing up, this study contributes to the SciTS by introducing Dynamic Networking (DN) concept that refers to some aspects of academic relationships that were largely neglected in prior literature and it provides the empirical evidence that in the context of non-Anglophone business science in Poland DN leverages academic productivity, while it is leveraged itself by the scholars' English language skills. Answering the call in SciTS to investigate academic collaboration rather from the perspective of individual scholars and their behaviour than from the perspective of team behaviour and team effects and to identify tools helpful in mitigating threats of collaboration [6, 9, 10], this study conceptualizes DN at the level of behaviour of an individual scholar that helps the focal scholars to mitigate the risks related to collaboration dynamics, specifically through attempts to enlarge academic network and critically evaluate existing connections. Our research addresses also recent SciTS call *to potentially integrate approaches, theories, and methods from across disciplines* [9, p. 543], because in developing and interpreting our research with DN as a focal construct we have combined existing theories of SciTS with strategy research theories, i.e. dynamic capabilities view and linguistic theories, i.e. critical period hypothesis.

## Conclusions

### Practical implications

Our research has practical value in terms of recommendations for various actors of academic systems, especially for non-Anglophone academic context. Firstly, this study encourages scholars for applying a selective approach towards academic networking: systematic seeking new

connection and focusing on partners that in a given period of time may be beneficial in terms of unique expertise and experience they bring. Such approach may be to some extent achieved through specific social platforms, e.g. finding new contacts within ResearchGate or LinkedIn), however traditional methods are relevant as well (i.e. meeting new people at conferences). Secondly, managers at universities (e.g. HR managers and top managers) could implement some special programmes for local faculties that would prepare them for initiating new contacts and networking, especially on an international level. Such programmes should be especially oriented at contacts with well-published scholars, e.g. scholars employed at recognizable academic institutions, but they should also help scholars to engage in collaborations with most productive scholars at their home faculties. Thus, we believe that universities should employ not only training programmes but also facilitate team spirit among employees, e.g. funding grants for inter-departmental projects. The training programmes are needed especially for developing language skills, where all four language skills of writing, speaking, reading and listening should be practiced. Central administration on country or international level (e.g. European Commission) should focus on special programmes financing short-term and long-term visits of scholars at foreign research institutions aimed at collaboration between less experienced and well experienced, well-published scholars. Obviously, such programmes exist to some extent (e.g. EU Marie Curie fellowships, etc.), but they are often oriented at junior faculty (e.g. for people no older than few years after doctorate), while there are many middle-aged and older scholars from post-communist countries that could definitely benefit from such programmes as they did not manage to develop international contacts and language capabilities at earlier stages of their careers.

## Limitations and further research

The research has some advantages and disadvantages as well. The sample of scholars was quite similar to other survey-based studies conducted in the context of one country and/or one discipline [46, 80, 106], but, obviously, it is much smaller than in prior studies that applied fully bibliometric approach, i.e. using pre-existing bibliometric databases only [22]. Using self-reported measures of academic productivity and other related constructs may be treated as a disadvantage, but in case of underdeveloped academic systems, like the Polish one, survey-based data appears to be a reasonable alternative [46]. Using cross-functional data and single-informant approach is always problematic, especially when it comes to the risk of social desirability phenomenon [107]. We controlled this problem using various ex-ante and ex-post procedures to mitigate and test for common method variance. Additionally, the results of study 1, where connections between all independent variables and dependent variable were estimated with the use of the same data source, we have also cross-validated these results through re-analysing the hypothesized connections using Scopus-based measures for research productivity (study 2). Although the sample we used to cross-validate our results was relatively small fraction of general sample (n = 73), this fraction is reasonable considering the lack of available data in non-Anglophone research contexts [46]. We argue that study 2 provides strong cross-validation of study 1 results, because in contrast to study 1, study 2 uses publication data only with regard to Scopus-based English language journal publications. Thus, study 2 uses productivity measure that is more aligned to the measures typical for Publish or Perish international academic standards. We believe that future research can further validate our research as we observe the dynamic increase in the visibility of Polish business scholars in Scopus database between 2014, where study 1 was conducted and 2020 (study 2). Thus, further research would probably enable validating our research results on a larger sample of Polish scholars with the use of Scopus data.

Our study is also restricted in terms of using one specific measure of academic productivity, i.e. "normal count" or the number of publications authored by the given scholar [10], instead of using "fractional count", i.e. number of publications divided by the number of authors [10], whereas the latter was found as a valuable measure of collaboration output, because few studies reported no link between collaboration/co-authorship and such productivity defined this way [106, 108]. Although future research should validate if DN as a focal construct introduced here is truly impactful with regard to other measures of productivity, we do not think that not-using fractional count disqualifies our research results. First of all, "the normal count" still seems to be the most popular productivity measure in SciTS [9]. Secondly, there are actually much more available measures than just "fractional" vs. "normal" publication count as academic productivity can be also measured through such factors as high-impact articles and citations [106, 109], research grants applications, novelty attributed to patents [9] and scholars' career advancement [6]. It seems impossible to include all of these productivity measures in one research project, so such limitation is an attribute of every study in SciTS. Recently [6], concluded in their extensive review of SciTS that there is a need for "*more careful measurement of impacts as opposed to outputs*" (p. 1). We argue that our main contribution to SciTS is in conceptualizing DN as a distinct collaboration-related research concept and providing valid and reliable survey-based measurement for this concept.

There are some general directions for further research. First of all, it would be very interesting to use larger dataset including scholars from various scientific disciplines and compare the importance of dynamic networking across such disciplines in the same fashion that appeared to be very fruitful for empirical research by [22]. Applying social network analysis (SNA) tools to data concerning networking scholars would allow us for a better control of the structural aspects of networking. For example, it would be interesting to see, to what extent the dynamic networking as a communication pattern results in enlarging centrality of the focal scholar within the network. We see also potential in testing validity of dynamic networking construct on the sample of scholars from most developed academic institutions (e.g. USA or UK). Logically, in such contexts it would be most interesting to see if DN impacts research productivity, while controlling for the impact of some other forms of research collaboration, however there is also potential to test there some other antecedents of DN as English language skills are "natural antecedent" in non-Anglophone context only. However, DN may appear to be a fruitful behavioural pattern for scholars from richest Anglophone countries, especially that in comparison to their peers from less developed non-Anglophone regions, they are more likely to function as "brokers" in international academic networks, because their institution may attract many candidates for PhD studies from developing countries. It was also found that "tactician" and "mentoring" networking style" is beneficial for them [110] and in general that the collaborations they build are characterized by strong pragmatism [111, 112]. As the first, intuitive step, this study would also benefit from its validation in other non-Anglophone countries of the similar level of socio-economic development like Poland, e.g. Slovenia, Greece, Portugal and Italy [113–115] (Last but not least, future research could apply multi-dimensional definition of academic productivity, comprising to larger extent other outputs measured on the level of individual scholar, such as citations, teaching scores and success rate with regard to grant applications.

## Supporting information

**S1 Appendix.**
(DOCX)

**S1 File.**
(ZIP)

## Author Contributions

**Conceptualization:** Anna L. Wieczorek, Maciej Mitręga, Vojtěch Spáčil.

**Data curation:** Anna L. Wieczorek, Maciej Mitręga.

**Formal analysis:** Maciej Mitręga, Vojtěch Spáčil.

**Funding acquisition:** Maciej Mitręga.

**Investigation:** Anna L. Wieczorek, Maciej Mitręga.

**Methodology:** Anna L. Wieczorek, Maciej Mitręga, Vojtěch Spáčil.

**Project administration:** Anna L. Wieczorek, Maciej Mitręga.

**Resources:** Anna L. Wieczorek, Maciej Mitręga.

**Software:** Maciej Mitręga.

**Supervision:** Maciej Mitręga.

**Validation:** Maciej Mitręga.

**Visualization:** Maciej Mitręga.

**Writing – original draft:** Maciej Mitręga, Vojtěch Spáčil.

**Writing – review & editing:** Anna L. Wieczorek, Maciej Mitręga, Vojtěch Spáčil.

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
