## [Editor Report · Decision Letter 0]

30 Apr 2020

PONE-D-20-10186

Networking and English language to sustain career in academic community under transition

PLOS ONE

Dear Dr. Wieczorek,

Thank you for submitting your manuscript to PLOS ONE. After careful consideration, we feel that it has merit but does not fully meet PLOS ONE’s publication criteria as it currently stands. Therefore, we invite you to submit a revised version of the manuscript that addresses the points raised during the review process.

Before sending our manuscript out for review, I would like to ask you to revise some formal aspects of the text:

- Text structure: Please, try to re-organise your text according to PLOS ONE's guidelines (see https://journals.plos.org/plosone/s/submission-guidelines#loc-manuscript-organization). Currently, the boundaries between the 'Materials and Methods' section with the Introduction and the Results sections are blurred. This makes the reading more difficult, as references to previous works in the literature, justification of the applied methods and presentation of the obtained results are mixed.

- Methodology justification: The applied methods should be properly presented and justified. For example, what were the criteria to select the potential responders to be invited to take the survey (i.e. all the scholars above 40 years old)? How were the questions of the survey designed (e.g. why productivity is measured by blind-reviewed journals)? In relation to this, notice PLOS' publication criterion #3 (https://journals.plos.org/plosone/s/criteria-for-publication#loc-3).

We would appreciate receiving your revised manuscript by Jun 14 2020 11:59PM. To enhance the reproducibility of your results, we recommend that if applicable you deposit your laboratory protocols in protocols.io, where a protocol can be assigned its own identifier (DOI) such that it can be cited independently in the future. For instructions see: http://journals.plos.org/plosone/s/submission-guidelines#loc-laboratory-protocols

We look forward to receiving your revised manuscript.

Kind regards,

Sergi Lozano

Academic Editor

PLOS ONE

Journal Requirements:

2. In your Methods section, please provide additional information about the participant recruitment method and the demographic details of your participants. Please ensure you have provided sufficient details to replicate the analyses such as: a) the recruitment date range (month and year),  b) a description of how participants were recruited, and c) descriptions of where participants were recruited and where the research took place, including the names of the universities where the participates were surveyed.

3. Please also include additional information regarding the survey or questionnaire used in the study and ensure that you have provided sufficient details that others could replicate the analyses. For instance, if you developed a questionnaire as part of this study and it is not under a copyright more restrictive than CC-BY, please include a copy, in both the original language and English, as Supporting Information.

5. We note you have included a table to which you do not refer in the text of your manuscript. Please ensure that you refer to Table 1 and 2 in your text; if accepted, production will need this reference to link the reader to the Table.
---

## [Author Response · Author response to Decision Letter 0]

16 May 2020

Dear Reviewers,

thank you for your comments and suggestions provided. We tried to address all of them and they are uploaded (together with your comments) in a separate file called Response to Reviewers.

---

## [Decision Letter · Decision Letter 1]

14 Jul 2020

PONE-D-20-10186R1

Networking and English language to sustain career in academic community under transition

PLOS ONE

Dear Dr. Wieczorek,

Thank you for submitting your manuscript to PLOS ONE. After careful consideration, we feel that it has merit but does not fully meet PLOS ONE’s publication criteria as it currently stands. Therefore, we invite you to submit a revised version of the manuscript that addresses the points raised during the review process.

As you can see below, both reviewers found the topic of your work interesting and relevant. However, they also raised a number of serious concerns. In particular, your revision should especially address the following issues:

- Work positioning in the state-of-the-art: Both reviewers asked to develop further the literature review. As stated by Reviewer 1, this is needed to precisely identify the gap the work aims to address and, eventually, frame your conclusions (see his/her comments 1) and 5) ).

- Methodological issues: Both reviewers raised methodological concerns (see comments 3 - 5 by Reviewer 1, and comment 3 by reviewer 2). Notice that PLOS publication criterion #3: Experiments, statistics, and other analyses are performed to a high technical standard and are described in sufficient detail." (https://journals.plos.org/plosone/s/criteria-for-publication#loc-3).

We look forward to receiving your revised manuscript.

Kind regards,

Sergi Lozano

Academic Editor

PLOS ONE

Reviewers' comments:

Reviewer's Responses to Questions

**Comments to the Author**

1. If the authors have adequately addressed your comments raised in a previous round of review and you feel that this manuscript is now acceptable for publication, you may indicate that here to bypass the “Comments to the Author” section, enter your conflict of interest statement in the “Confidential to Editor” section, and submit your "Accept" recommendation.

Reviewer #1: (No Response)

Reviewer #2: All comments have been addressed

2. Is the manuscript technically sound, and do the data support the conclusions?

Reviewer #1: No

Reviewer #2: Yes

3. Has the statistical analysis been performed appropriately and rigorously? 

Reviewer #1: No

Reviewer #2: Yes

4. Have the authors made all data underlying the findings in their manuscript fully available?

Reviewer #1: Yes

Reviewer #2: Yes

5. Is the manuscript presented in an intelligible fashion and written in standard English?

Reviewer #1: Yes

Reviewer #2: Yes

6. Review Comments to the Author

Reviewer #1: The article concerns an important topic. Important for theoretical but also practical reasons. It is also valuable that it concerns the functioning of science outside the leading countries in terms of quantity and quality of scientific production. However, one can have numerous, serious reservations about the article in its present shape. Here are some key issues:

1)

The authors emphasize that “prior studies have generally investigated academic collaboration as a “black box” and “Prior research treated academic collaboration usually a ‘black box’ focusing mainly on co-authorship while academic networking is a complex”. This statement is entirely false. It is enough to mention the recent literature reviews about "team science", in which numerous examples of unpacking the "black box" are cited:

• Hall, K. L., Vogel, A. L., Huang, G. C., Serrano, K. J., Rice, E. L., Tsakraklides, S. P., & Fiore, S. M. (2018). The science of team science: A review of the empirical evidence and research gaps on collaboration in science. American Psychologist, 73(4), 532.

• Leahey, E. (2016). From sole investigator to team scientist: Trends in the practice and study of research collaboration. Annual review of sociology, 42, 81-100.

• Sonnenwald DH (2007) Scientific collaboration. Ann. Rev. Info. Sci. Tech. 41 (1): 643–681.

Or classical papers, as Katz JS, Martin BR (1997) What is research collaboration. Research Policy 26 (1): 1–18.

What is more, scientific cooperation studies using randomized trials have recently started to appear. For example:

• Boudreau, K., Ganguli Prokopovych, I., Gaule, P., Guinan, E. C., & Lakhani, K. R. (2012). Colocation and scientific collaboration: evidence from a field experiment. Harvard Business School working paper series# 13-023.

In this case, it is not just about the pair of missing footnotes. The problem is more serious. Without a reference to the current state of knowledge and ongoing discussions in the field, it is difficult to assess what contribution the reviewed article would make to the development of the discipline. This aspect should be carefully thought out by the authors and clearly explained in the text.

2)

The research procedure, especially sampling, seems to make it impossible to draw conclusions that the authors want to draw. Anyway, the authors admit it themselves, but they draw an optimistic conclusion that in total this approach does not disqualify the study:

“Thus, we can assume that the final sample was skewed with regard to the normal distribution of research productivity, i.e. we selected more productive scholars than the average scholar, but that was not treated as an issue because we did not aim at representative sampling, but instead we wanted to test hypothesized model on the purposively selected sample of Polish academicians that are likely active in publishing and networking as well.”

I must admit that I do not understand this argument. It seems that since we deliberately choose the most productive researchers, it will be difficult to assess where productivity comes from. A similar caveat can be found in relation to the networking variable - since respondents were asked to send a survey to friends, it can be expected that it went proportionally more often to more networked people.

3)

I understand reluctance regarding bibliometric indicators:

“This study follows [22], who suggest bibliometric techniques as irrelevant for investigating scholars from less developed countries, because such scholars are not well represented in international scientific databases.”

However, ultimately the authors use the bibliometric variable. The only difference is that it comes from a survey. It may, therefore, be imprecise. In the case of a sample of about 200 people, one could be tempted to manually check the list of publications (e.g. based on CVs, websites of the institutions, etc.), and the national publication database (as far as I know there is one in Poland, so you do not have to limit yourself to international sources, which in the case of non-English-speaking countries give a distorted picture of scientific production).

Adding an objective indicator of publication output would also allow controlling for co-authorship and the number of co-authors. It is essential because it can be expected that the way of counting the number of publications can significantly influence the results of the analysis (whole counting vs fractional counting).

4)

It seems to me that the methodological approach is not sufficiently justified. It is not clear why the authors immediately switch to a rather complex SEM-based approach. Wouldn't a more straightforward approach using maximum likelihood estimation for count variables (e.g. Poisson regression, negative binomial, etc.) be enough?

At a more detailed level, two potential problems can be identified. First, why the gender variable is not included in the main model but is only used in the additional analysis?

Secondly, it is worth considering the possible correlation of responses of individual people working in the same institutions. Did the authors include these possible correlations in the model? (For example, creating a multi-level model or adding dummy variables for individual institutions). Based on the reviewed text, it seems not. As a result, it is difficult to assess whether the values of the coefficients and the reported standard errors are reliable.

5)

The conclusions, and especially the recommendations given by the authors, seem quite clichéd, it is difficult to see the value of novelty in them. We know that cooperation, in its various dimensions, translates into the productivity of scientists. We know that knowledge of English helps in publishing. We know that we should support the development of cooperation. I'd instead like to see narrow conclusions in this section, but with something new.

6)

Finally, a small note. I have doubts about the correctness of the use of the term “lingua franca” by the authors. Dictionary definition says that “lingua franca: a language that is adopted as a common language between speakers whose native languages are different”. While the authors write:

“In the last two decades Polish academic system has gradually become more “open” (e.g. increasing participation in international conferences and top-tier journals), however, Polish language has remained lingua franca for Polish scholars (e.g. the tradition of PhD thesis writing in Polish language)”.

Bearing in mind the definition quoted, it seems that the native language spoken mostly in one country cannot be lingua franca for its native speakers.

Reviewer #2: An interesting paper that sheds light on the networking patterns in a non-English academic community. The text is well written and clear in terms of argumentation line and hypotheses testing. I suggest accepting this paper with some minor clarifications as follows.

1. If possible, please provide some correspondence between your sample and available statistics on human resources in science and technology in Poland as it may help to understand which particular groups you represent in your study.

2. It would be great to see some references to similar or related studies from other posts-soviet countries you are mentioning in your analysis if you know any.

3. I still have some doubts about the use of bootstrapping to increase your sample size from 198 to 5000 cases. Do you have any ways to assess trace fraction or bootstrapping was used only for probability distributions, and you don't try to represent any particular social group from the population?

7. PLOS authors have the option to publish the peer review history of their article (what does this mean?). If published, this will include your full peer review and any attached files.

Reviewer #1: No

Reviewer #2: No

---

## [Author Response · Author response to Decision Letter 1]

17 Aug 2020

Due to its lenghth, response to Reviewers is a separate file attached.

---

## [Decision Letter · Decision Letter 2]

11 Nov 2020

PONE-D-20-10186R2

Dynamic Academic Networking Concept and its Links with English Language Skills and Research Productivity - non-Anglophone Context

PLOS ONE

Dear Dr. Wieczorek,

Thank you for submitting your manuscript to PLOS ONE. After careful consideration, we feel that it has merit but does not fully meet PLOS ONE’s publication criteria as it currently stands. Therefore, we invite you to submit a revised version of the manuscript that addresses the points raised during the review process.

Specifically, I would like to ask you to address the comments and clarification requirements made by Reviewer 2, as well as minor points raised by Reviewer 3.

We look forward to receiving your revised manuscript.

Kind regards,

Sergi Lozano

Academic Editor

PLOS ONE

Reviewers' comments:

Reviewer's Responses to Questions

**Comments to the Author**

1. If the authors have adequately addressed your comments raised in a previous round of review and you feel that this manuscript is now acceptable for publication, you may indicate that here to bypass the “Comments to the Author” section, enter your conflict of interest statement in the “Confidential to Editor” section, and submit your "Accept" recommendation.

Reviewer #2: All comments have been addressed

Reviewer #3: All comments have been addressed

2. Is the manuscript technically sound, and do the data support the conclusions?

Reviewer #2: Yes

Reviewer #3: Yes

3. Has the statistical analysis been performed appropriately and rigorously? 

Reviewer #2: Yes

Reviewer #3: Yes

4. Have the authors made all data underlying the findings in their manuscript fully available?

Reviewer #2: Yes

Reviewer #3: Yes

5. Is the manuscript presented in an intelligible fashion and written in standard English?

Reviewer #2: Yes

Reviewer #3: Yes

6. Review Comments to the Author

Reviewer #2: Dear authors,

Thank you for addressing my early comments and revising your manuscript in line with the recommendations of both reviewers. I read the text carefully and found your paper much more consistent. However, due to rewriting a few points appeared I want to draw your attention.

At page 1 you say: «Logically, in comparison to other disciplines team social science is also very risky because social science research methods are less standardized and the researchers’ cultural backgrounds are more meaningful due to interpretative epistemology bringing more potential for team conflicts». Thought this argument seems reasonable it doesn’t come from the observation that «collaborative research projects in social science are much longer [8].» Please either provide some evidence for this or reconsider your statement in a more neutral manner.

You pay attention to the importance of English language in collaboration, but do not specify whether this collaboration should be international (should it be?). I am also not sure that your example of science-industry collaboration (see page 17) is relevant to the topic of your research as it might take place with local producers.

The section on English skills and dynamic networking can be shortened and more focused on the arguments why English matters (other languages may also be important) instead of providing a list of non-Anglophone countries :)

At page 23 you mention scholars that may have «have strongest networks and easiest access to resources» and call them «brokers». I find the term «gatekeeper» would be more suitable for such a definition as it refers to the network control function rather than to the service provision. I might haven’t got your point here.

I don’t have questions about Study 1, but for Study 2 I support the other reviewer’s comment on the necessity of data verification& I suggest looking not only at Scopus which is English language bias but also at other open sources that can provide additional information about publication activity of Polish authors, e.g. ResearchGate, Academia.edu. Even Google Scholar can be relevant here as long as you try to assess the overall progress of particular authors and make a longitude of it. Otherwise provide stronger arguments why Scopus remains the only relevant source for your study.

I still don’t understand why connections with industry should have an effect on publication activity. This line of argumentation remains unclear to me.

Reviewer #3: The paper is clear and argumentative. The fact that worldwide research is English-based is an interesting subject of research, especially when can help scholars to reflect on their career. The historical contextualization in scholars that have been affected by the presence of the Berlin Wall is relevant in a context where English is now required to spread research. It is also pleasant to see references on European policies at this regard, as international mobility is a valuable form of both cultural and linguistic integration. Employed methods shows clearly the impact of past education in Poland, which should recognize the role of EU in the last decades. In that sense, it would be interesting to employ the same methods to analyze other languages that in the future can be predominant in research environments, such as Chinese, Portuguese, and Indians, e.g. I simply say that the method can be further validated with a different point of view. Thanks for the pleasant reading.

P.S. Please notice that Figure 1 is missing and many double spaces were found during the reading. I suggest a simple review with a tool like Grammarly.

7. PLOS authors have the option to publish the peer review history of their article (what does this mean?). If published, this will include your full peer review and any attached files.

Reviewer #2: No

Reviewer #3: **Yes: **Dario Rodighiero

---

## [Author Response · Author response to Decision Letter 2]

23 Dec 2020

Reviewer #2

Thank you for addressing my early comments and revising your manuscript in line with the recommendations of both reviewers. I read the text carefully and found your paper much more consistent. However, due to rewriting a few points appeared I want to draw your attention.

At page 1 you say: «Logically, in comparison to other disciplines team social science is also very risky because social science research methods are less standardized and the researchers’ cultural backgrounds are more meaningful due to interpretative epistemology bringing more potential for team conflicts». Thought this argument seems reasonable it doesn’t come from the observation that «collaborative research projects in social science are much longer [8].» Please either provide some evidence for this or reconsider your statement in a more neutral manner.

First, thank you very much for appreciating our revision. 

Thank you also for spotting a blurred logical sequence in two sentences in the introduction. It was improved in the current version.

You pay attention to the importance of English language in collaboration, but do not specify whether this collaboration should be international (should it be?). I am also not sure that your example of science industry collaboration (see page 17) is relevant to the topic of your research as it might take place with local producers.

Many thanks for your comments.

The main causal path (these that we focus on) are these presented in figure 1 (copied below). Only in the control variables we present and test the other connections, e.g. Corporate network -> Productivity; Home network -> Productivity. As we are introducing the new concept of “dynamic networking” to establish its nomological validity and also to assure that this concept is really distinct in relation to other concepts established in literature, we have included some other network-based control variables but we do not concentrate on them in our research. 

We are fully aware that the science - industry collaboration that we discuss in the paper (again with regard to control variables and to distinguish our focal construct from other constructs) may take local form only and we do not suggest in the manuscript that we focus on international science - industry collaboration. In the specific discipline and institutional context that our research is focused on, such international science - industry collaboration is extremely rare. We have focused on the population of scholars at the age of 40 or older in post-communist countries so concerning their habits and competences built largely in the communist time, they may have some ties with business practitioners but on local scale only and these ties were built mainly in the 90s., i.e. at the beginning of transformation period with fastest growing markets and these ties were used mainly as a secondary, very effective source of extra money for them, e.g. through commercial research projects. 

Similarly, the collaboration that we mainly focus on as represented by our focal construct of Dynamic Networking (see figure 1) is not about international networking only but it is rather about some activities that systematically expand work connections of focal scholars beyond their home faculty network (e.g. “I try systematically to broaden the area of my research contacts through initiating contacts with new people at national and international conferences”) and it is also about pragmatic approach towards existing network (e.g. “I systematically evaluate the usability of my current research contacts (e.g. taking into consideration such benefits as joint publications”). In the current version of the manuscript we give full consideration to these issues, including providing the definition of dynamic networking as routinized focal scholars’ patterns of dynamic interaction with other scholars that happen at various stages of collaboration cycle and are oriented at learning from others and purposively shaping network of a focal scholar.

Therefore, in our study we do not focus on “international networking’ but rather “dynamic networking” that goes beyond home network and includes, but is not limited to, international networks. We also make a distinction between network/collaboration on one hand and networking on the other in several parts in the manuscript. However, we still argue that English language is an important antecedent to dynamic networking even beyond networking on international scale, because language capabilities are the passage to good quality, effective team science nowadays, and specifically for the senior scholars in non-anglophone context. 

For example, the relevant part reads as follows:

Many inhabitants of non-Anglophone countries not only have worse command of English, but they also perceive the world differently which is caused by general cultural differences and also by the language they speak and by absence of some notions in their native language which, in turn, may make some phenomena difficult to understand for those people. Sapir–Whorf hypothesis [59, 60] explains differences in human cognition [61] and emotion [62] through differences in human language. In the context of academic interactions, it means that the way the scholar constructs his/her worldview is based on the language the scholar acquired and uses. Extrapolating this mechanism to the concept of DN, it could be that acquiring English language skills makes non-Anglophone scholars not only more capable of English-language based communication with other scholars, but also it may change their approach to interaction with other scholars. Specifically, as English language is perceived a language rich in pragmatics [63], the acquisition of English language may make a non-Anglophone scholar more pragmatic in interaction with other scholars, which enable DN and indirectly, it has positive effect on scholar’s productivity. Furthermore, taking into consideration that we build our understanding of the world through the lens of our native language, the lack of some words in a given language influences the way people think and act as its native users . We may take the notion of networking as an example of it. In Polish language there is no equivalent of the English word networking and many people simply do not know the word/concept at all. The notion is known by some business people, scientists or researchers who study this phenomenon, but they speak and write about it borrowing the English term, so one can assume they do not have a preunderstanding of the concept – those who do not do research on networking may even not know what networking really entails and as a result do not have proper abilities to network with others. In turn, a term collaboration which is closely connected with networking, but of course is not synonymous, has negative connotations for some Poles as it in their minds this notion may refer mainly to the traitors of the nation collaborating with Germans at the times of II World War (such WWII practices were called collaboration, “kolaboracja” in Polish). This can make some scholars associate the concept of collaboration with unethical cooperation practices (like e.g. ghostwriting) and limits their understanding of the wide spectrum of networking actions that are available, i.e. initiating new connections with scholars for mutual learning.

The section on English skills and dynamic networking can be shortened and more focused on the arguments why English matters (other languages may also be important) instead of providing a list of non Anglophone countries :)

Thank you for this comment, we believe that following the course of action you suggested will make the section more logical. Indeed, we tried to shorten the text wherever possible – for instance we deleted the list of non-Anglophone countries and tried to once again explain why English language is considered a global language now (in general and as a global language of science). It was not, however, possible to shorten the further parts of the section (i.e. the part where we explain the context of post-communist countries and the impact of communist times on linguistic skills of senior scholars) as we felt that we should explain it in detail why people from such countries may be disadvantaged linguistically due to post-communist heritage of their countries of origin and living. We also felt it was necessary not to shorten the part referring to Sapir-Whorf hypothesis as it explains how a language we speak influences the way we perceive the world and how we think generally and we present there how it could have impacted the attitudes of post-communist scholars towards e.g. networking.

At page 23 you mention scholars that may have «have strongest networks and easiest access to resources» and call them «brokers». I find the term «gatekeeper» would be more suitable for such a definition as it refers to the network control function rather than to the service provision. I might haven’t got your point here.

Ok, thank you. We have changed terms here.

I don’t have questions about Study 1, but for Study 2 I support the other reviewer’s comment on the necessity of data verification& I suggest looking not only at Scopus which is English language bias but also at other open sources that can provide additional information about publication activity of Polish authors, e.g. ResearchGate, Academia.edu. Even Google Scholar can be relevant here as long as you try to assess the overall progress of particular authors and make a longitude of it. Otherwise provide stronger arguments why Scopus remains the only relevant source for your study.

Thank you. We understand your concern. Actually it is difficult to find the balance between or even combine the Polish post-communist academic system with typical Western Anglophone academic systems, where top-tier journal publications are almost only in centre of attention when it comes to research productivity measurement. However, we would like to emphasize that Polish academic system, including Polish business science where our study is located, underwent dynamic changes in the last decade, which made it much closer to Western systems of productivity measurement. Our study is based on data gathered at different points of time for the dependent variable: 2014 (study 1) and 2020 (study 2). For study 1, the wide focus on productivity measured through publications of various kind is understandable, while the focus on only Scopus registered publications in the study 2 was also a very conscious choice considering the institutional changes observed in last decade. To explain our rational we have extended relevant part of the manuscript and this part reads as follows now:

The pace of changes of Polish academic system, similarly to other post-communist systems of CEE (Central and Eastern Europe) has substantially progressed in the last decade. Analysing the publication patterns of non-English-speaking European countries: Czech Republic, Denmark, Finland, Flanders (Belgium), Norway, Poland, Slovakia, and Slovenia, [81] found that in the period between 2011 and 2014 the publication patterns were stable in West European and Nordic countries, whereas CEE countries experienced considerable changes towards more and more emphasis in English language publications in international journals. For Poland as the biggest CEE country the trend continues with regard to impact factor publications which is largely based on the strong impact of policy instruments with regard to country’s national research evaluation system, academic promotion procedures and competitive grants [82]. In October 2018 the new Law on Higher Education and Science was established in Poland (also called Science Constitution 2.0 to emphasize the depth of institutional changes) which centralized leadership of the university authorities [83] and at the same time imposed much more demanding measures of scientific productivity based on publication outputs in Scopus database. Specifically, the new ranking of publications in journals was mostly based on SNIP (The Source-Normalized Impact per Paper) indicator which is a relatively new journal metric by Elsevier's Scopus [84]. The SNIP was selected because this indicator acknowledges citation potential related to differences in topicality across scientific fields. Therefore, it may be concluded that in contrast to previous periods, nowadays the research productivity of business scholars in Poland is evaluated mostly through their publications registered in Scopus. This institutional change motivated Polish scholars to publish more and more in highly ranked journals, so it may be assumed that the trend identified for the period 2011-2016 [81, 82] received some additional impetus. It translates into even more challenging situation for the older generation of Polish scholars who started their careers in communist period, where publications in Polish language were the main pattern of dissemination of their research results. 

I still don’t understand why connections with industry should have an effect on publication activity. This line of argumentation remains unclear to me

Thank you. Again, please notice that the academic-industry ties have been introduced into the model as control variable only, so it means that the focal causal path does not include any influence from such ties on publication activity. We have actually presented the line of argumentation explaining that existing academic-industry ties do not impact on publications effects due to specific context of our research (post-communist “experienced scholars” with some business connections that are utilized mostly as source of extra income only). However, we have included this aspect as a control variable because prior research has tested this connection and found some empirical support for it. Following your concern the relevant section related to control variables was slightly revised and it reads as follows now:

In turn, Corporate network corresponds with University-industry collaboration concept [8] from the perspective of an individual scholar and refers to professional relationships between focal scholar and business practitioners. This specific cross-industry collaboration was perceived as a driver of publication productivity in prior research, because it increases the innovativeness of academic research [85, 86], however as explained before in this study, it is treated as a control variable only, due to specific the institutional context of our research (see previous section).

 

Reviewer #3: 

The paper is clear and argumentative. The fact that worldwide research is English based is an interesting subject of research, especially when can help scholars to reflect on their career. The historical contextualization in scholars that have been affected by the presence of the Berlin Wall is relevant in a context where English is now required to spread research. It is also pleasant to see references on European policies at this regard, as international mobility is a valuable form of both cultural and linguistic integration. Employed methods shows clearly the impact of past education in Poland, which should recognize the role of EU in the last decades. In that sense, it would be interesting to employ the same methods to analyze other languages that in the future can be predominant in research environments, such as Chinese, Portuguese, and Indians, e.g. I simply say that the method can be further validated with a different point of view. Thanks for the pleasant reading.

Dear Reviewer, thank you so much for your kind attitude towards our study. We totally agree that there are many ways that our findings may be validated and expanded in further research. We have proposed several further research directions in the last section of our manuscript. 

When it comes to Figure 1 which you claimed was missing, we double-checked and it was uploaded to the system. It was, however, uploaded in eps format (according to PLoS instructions) and maybe for that reason you could not see it. We copied the figure to this file (see above, p. 2).

---

## [Decision Letter · Decision Letter 3]

12 Jan 2021

Dynamic Academic Networking Concept and its Links with English Language Skills and Research Productivity - non-Anglophone Context

PONE-D-20-10186R3

Dear Dr. Wieczorek,

We’re pleased to inform you that your manuscript has been judged scientifically suitable for publication and will be formally accepted for publication once it meets all outstanding technical requirements.

Kind regards,

Sergi Lozano

Academic Editor

PLOS ONE

Additional Editor Comments (optional):

Dear authors,

after checking the report by Reviewer 2, I have decided to accept the article for publication. Nevertheless, please notice that there is a typo right before the very last sentence of the text, as well as some format errors at the reference lists.

Sergi.

Reviewers' comments:

Reviewer's Responses to Questions

**Comments to the Author**

1. If the authors have adequately addressed your comments raised in a previous round of review and you feel that this manuscript is now acceptable for publication, you may indicate that here to bypass the “Comments to the Author” section, enter your conflict of interest statement in the “Confidential to Editor” section, and submit your "Accept" recommendation.

Reviewer #2: All comments have been addressed

2. Is the manuscript technically sound, and do the data support the conclusions?

Reviewer #2: Yes

3. Has the statistical analysis been performed appropriately and rigorously? 

Reviewer #2: Yes

4. Have the authors made all data underlying the findings in their manuscript fully available?

Reviewer #2: Yes

5. Is the manuscript presented in an intelligible fashion and written in standard English?

Reviewer #2: Yes

6. Review Comments to the Author

Reviewer #2: The paper is clear and argumentative, the data support the conclusions. All comments have been addressed. Thanks for the pleasant reading.

7. PLOS authors have the option to publish the peer review history of their article (what does this mean?). If published, this will include your full peer review and any attached files.

Reviewer #2: No

---

## [Editor Report · Acceptance letter]

22 Jan 2021

PONE-D-20-10186R3 

Dynamic Academic Networking Concept and its links with English Language Skills and Research Productivity –non-Anglophone context 

Dear Dr. Wieczorek:

I'm pleased to inform you that your manuscript has been deemed suitable for publication in PLOS ONE. Congratulations! Your manuscript is now with our production department. 

Kind regards, 

on behalf of

Dr. Sergi Lozano 

Academic Editor

PLOS ONE